# The Influencing Factors of HIV-Preventive Behavior Based on Health Belief Model among HIV-Negative MSMs in Western China: A Structural Equation Modeling Analysis

**DOI:** 10.3390/ijerph191610185

**Published:** 2022-08-17

**Authors:** Hui Liu, Guichuan Lai, Guiqian Shi, Xiaoni Zhong

**Affiliations:** Department of Epidemiology and Health Statistics, School of Public Health, Chongqing Medical University, Yixue Road, Chongqing 400016, China

**Keywords:** health belief model (HBM), pre-exposure prophylaxis (PrEP) adherence, men who have sex with men (MSM), HIV-preventive behavior, western China

## Abstract

(1) Background: Men who have sex with other men (MSMs) are at high risk of being infected by the human immunodeficiency virus (HIV) in western China. Pre-exposure prophylaxis (PrEP) is an efficient way to prevent HIV transmission. However, adherence is the most vital determinant factor affecting PrEP effectiveness. We conducted a study based on the Health Belief Model to explore factors that predict adherence to PrEP among a cohort of 689 MSMs in western China. (2) Methods: We assessed perceived susceptibility, severity, benefits, barriers, self-efficacy, cues to action, and HIV-preventive behavior through a cross-sectional survey. (3) Results: PrEP self-efficacy was directly associated with PrEP behaviors (β = 0.221, *p* < 0.001), cues to action were directly associated with PrEP behaviors (β = 0.112, *p* < 0.001), perceived benefits were directly associated with PrEP behaviors (β = 0.101, *p* < 0.001), and perceived susceptibility was directly associated with PrEP behaviors (β = 0.117, *p* = 0.043). (4) Conclusion: Medication self-efficacy, perceived susceptibility, and cue to action structures are predictors of the MSMs’ HIV-preventive behavior in western China. These results will provide theoretical plans for promoting PrEP adherence in MSMs.

## 1. Introduction

The human immunodeficiency virus (HIV) is a major global public health problem. China has the second-largest number of people living with HIV in Asia, accounting for 1.29 million [1]. In recent years, the acquired immune deficiency syndrome (AIDS) epidemic in China has grown rapidly from 440,000 in 2011 to 1,050,000 in 2021 [2]. Men who have sex with men (MSMs) constitute the main key population of HIV-infected individuals, and over 95% of HIV infections in China were sexually transmitted [3]. The risk of HIV infection in MSMs is about 20 times that in the general population on average [4], and more than 25% of MSMs comprise new HIV infection cases in China despite various interventions having been adopted [5,6]. Western China has the highest number of HIV/AIDS new infections in China, accounting for 45% of new HIV/AIDS cases in China [7]. Therefore, the MSM population should be the focus of strategies for preventing and controlling AIDS.

Highly effective protection against HIV infections in MSMs has been affirmed in the pre-exposure prophylaxis (PrEP) trials [8,9,10]. However, the effectiveness of PrEP largely depends on medication adherence according to the ninth International AIDS Society conference (WHO, 2017) on HIV science in 2017 and our previous research [11,12,13]. Adherence may be the most vital determinant factor affecting PrEP effectiveness, and improving adherence has become a common topic accompanied by evidence from existing studies [4].

Medication adherence is a preventive behavior dependent on behavioral decision theory [14,15,16]. Determining the main factors affecting preventive behavior is necessary to improving PrEP adherence in MSMs, but few studies have explored the predictive factors of adherence to PrEP or HIV-preventive behavior in MSMs in western China.

The study was guided by the Health Belief Model (HBM), which explains the relationship between health behaviors and psychological factors. The HBM has been successfully applied to predicting health behaviors [17,18,19,20,21,22] but has been rarely used in studying PrEP medication adherence in HIV-negative MSM populations. Therefore, this study analyzed the HIV-preventive behavior of HIV-negative MSMs in western China from a new perspective.

## 2. The HBM Model

The HBM was first proposed by American social psychologists Godfrey Hochbaum and Irwin Rosenstock in the 1950s [23]. It is a critical theory that explains how individual health behavior is affected by psychological factors and mainly emphasizes that the dominant element of individual health behavior is a personal subjective belief [23,24]. The HBM has been adopted as a conceptual framework and extensively evaluated empirically. It can explain health-promoting behavior and the association between health behaviors and psychological activities [25,26]. The basic HBM consists of six elements (Figure 1).

Perceived susceptibility refers to a person’s subjective perception of the risk of ac-quiring an illness or disease and is a powerful predictor of health behavior prevention in various populations [15,27,28,29]. Based on these studies, we established the following hypothesis:

**H1.** *Perceived susceptibility positively affects the preventive behaviors of MSMs*.

Perceived severity refers to a person’s subjective judgment of the severity of a poor health outcome, for example, disease. Some studies have shown that perceived severity is associated with health-related preventive behaviors, such as condom use and COVID-19 and STI prevention behaviors [28,30,31]. The more serious a disease is, the more preventive it may be. Therefore, we established the following research hypothesis:

**H2.** 
*Perceived severity has a positive impact on HIV-related prevention in MSMs.*


Perceived barriers refer to a person’s feelings about obstacles to performing a recommended health action [32,33] and as obstacles to behavior change, which negatively affects health-related behaviors. On the basis of the results of these studies, we established the following assumption:

**H3.** 
*Perceived barriers negatively affect HIV-preventive behaviors.*


Perceived benefits refer to a person’s perception of the effectiveness of various actions available to reduce the threat of illness or disease (or to cure illness or disease). These articles showed that perceived benefits promote changes in healthy behavior [34,35,36]. Therefore, we established the following hypothesis:

**H4.** 
*Perceived benefit positively affects the HIV-preventive behaviors.*


Cues to action are usually the critical “first step” in health action and can be internal (e.g., chest pains and wheezing) or external (e.g., advice from others, illness of families, and newspaper articles). Studies have shown that cues to action are factors for predicting health-related intentions and behaviors [28,35,37]. Based on previous studies, we make the following assumption:

**H5.** 
*Cues to action directly affect the HIV-preventive behaviors of MSMs.*


Self-efficacy refers to the level of a person’s confidence in his or her ability to perform a behavior successfully. The higher the self-efficacy is, the better a disease-preventive behavior is [38,39,40,41]. Accordingly, we made the following assumption about the prediction relationship between the self-efficacy and HIV-preventive behaviors of MSMs.

**H6.** 
*HIV-preventive behavior is positively affected by self-efficacy.*


According to the HBM, we hypothesized that preventive behavior is affected by perceived susceptibility, severity, perceived barriers, benefits, cues to action, and self-efficacy. Therefore, this study aimed to assess the factors associated with the HIV-related protective behaviors of MSMs in the southwest part of China.

## 3. Materials and Methods

### 3.1. Participants and Procedure

The participants in “The study on improving the compliance of pre-exposure prophylaxis in MSM based on intelligent reminder system (IRS) to reduce new HIV infections” (a cohort study of National Science and Technology Major Project in Western China from January 2018 to December 2021) were included (registration number: 2018ZX10721102-005). We recruited 1608 HIV-negative MSMs in western China by using a convenience sampling method. The volunteers were randomly divided into three groups: the IRS group (541 MSMs), the non-IRS group (544 MSMs), and the non-PrEP group (283 MSMs). In the IRS and non-IRS group, Lamivudine and Tenofovir tablets (300 mg + 300 mg) were taken orally daily. The inclusion criteria were as follows: (1) signed informed consent; (2) age between 18 and 65; (3) HIV antibody-negative; (4) intercourse with one or more male partners one month before the trial screening phases; (5) willingness to use the study medication under the guidance and to engage in follow-up arrangements; and (6) willingness to participate in the trial for 96 weeks. The exclusion criteria were as follows: (1) positive HIV antibody test result at the time of screening; (2) HBsAg (hepatitis B surface antigen) or anti-HBc (hepatitis B core antibody)-positive; (3) severe illness that the investigator believed to have affected the intervention, follow-up, or evaluation of the volunteers; (4) use of other study medications three months before the screening; (5) inability or unwillingness to provide informed consent or inability to comply with test requirements. The data collection process is shown in Figure 2. The method of self-administered questionnaire was adopted, and the questionnaires were collected on the spot and checked for completeness and logic [42,43]. Data were obtained in the first follow-up period for the cohort because HBM-related questions were only involved during this period.

### 3.2. Measurements

#### 3.2.1. Demographics

Demographic characteristics included age, household registration, ethnicity, educational level, employment status, occupation group, marital status, and disposable income.

#### 3.2.2. HBM Structural Factors

On the basis of our previous research [5,40], all questions used to describe factors that perceived susceptibility, severity, benefits, barriers, and medication self-efficacy were measured with five-point Likert questions. The measurement problem details of the six dimensions of HBM are shown in Table 1.

#### 3.2.3. Protective Behavior

The participants received free PrEP medication at the project’s clinic research center in each follow-up period. The follow-up questionnaire included questions about the number of times the subject took PrEP and missed doses during both follow-up periods.

Protective behaviors were identified using the PrEP medication rate, which ranges from 0 to 1. Thus, the medication rate in the study was determined using the formula (all daily PrEP medication) = (total number of pills prescribed for the follow-up period—number of missed doses)/total number of pills prescribed for the follow-up periods [1,5].

### 3.3. Data Analysis

The demographic characteristics of the MSMs were analyzed by using the SAS studio software (version 9.04; SAS Institute Inc., Cary, NC, USA). The reliability of the construct and structural model was evaluated using confirming factors analysis (CFA). The structural equation model (SEM) was used to verify the theoretical assumptions of the HBM model. Cronbach’s alpha was measured for the reliability test, and R2 was used to represent the explanatory degree of the independent variables concerning dependent variables. Weighted least-squares mean and variance-adjusted estimator was used in estimating the parameters. The fit of the measurement model was tested also through CFA. Five indices were used in assessing the model fit: comparative fit index (CFI), goodness of fit index (GFI), root mean square error (RMSEA), Beetle-Browed NFI, and chi-square normalized by degrees of freedom (χ^2^/df). According to classic research, CFI values greater than 0.94 suggest a good fit between data and hypothesized models [44]. RMSEA values less than 0.09 suggest a fair or adequate approximation error, whereas less than 0.055 suggest a small error [45]. The GFI should be 0.90 and above to present an acceptable fit with current data [44]. χ^2^/df should be less than three. The standards referenced by all indexes are generally accepted in statistics. A two-tailed and *p*-value < 0.05 were considered statistically significant.

## 4. Results

### 4.1. Sociodemographic Characteristic

A total of 689 MSMs were included in this analysis, and 46.59% were 25–34 years old, 71.82% lived in an urban household, 92.74% were of Han nationality, 71.55% had a college degree or above, 82.15% were employed, 76.68% were single, and 25.91% reported an income below RMB 3000, and 9.02% reported an income higher than RMB 10,000 per month.

### 4.2. Descriptive Analysis and Relationship between Constructs

The perceived severity of preventive behavior in MSMs had the highest mean score (mean = 4.48; SD = 0.711). The following mean scores were obtained: perceived benefits in terms of HIV prevention, 3.485 (SD = 1.421); medication self-efficacy, 2.832 (SD = 1.190); perceived barrier, 2.352 (SD = 1.299); and perceived susceptibility, 2.243 (SD = 1.061). Furthermore, the results of the descriptive analysis of MSMs’ HIV-preventive behaviors revealed that they had a moderate preventive behavior (median = 0.667; IQR = 0.9). Then, we examined the relationships among the seven constructs. A significant correlation was identified between HIV-preventive behavior and other constructs excluding perceived severity (Table 2).

### 4.3. Structural Model Results

In the study, the GFI, NFI, CFI, RMSEA, and χ^2^/df were 0.982, 0.965, 0.976, 0.037, and 1.954, respectively (Table 3). Thus, the goodness of fit of the structural model indicated an acceptable and satisfactory fit.

SEM analysis showed (Figure 3 and Table 1) that perceived severity, perceived barriers, and HBM constructs had been rejected and did not affect HIV-preventive behavior in MSMs. By contrast, self-efficacy (β = 0.221; *p* < 0.01) was the most important predictor of HIV-preventive behavior, followed by perceived susceptibility (β = 0.101; *p* < 0.01), perceived benefits (β = 0.117; *p* < 0.01), and cues to action (β = 0.112; *p* < 0.01). The path coefficients of perceived benefits and barriers to self-efficacy were 0.103 (*p* < 0.01) and −0.219 (*p* < 0.01), respectively. All pieces of information are shown in Figure 3 and Table 1.

## 5. Discussion

In December 2021, a total of 1.14 million HIV infection cases were reported nationwide in mainland China, accounting for 97% of sexual transmission cases. MSMs are at high risk of being infected with HIV or AIDS, as the estimated prevalence of HIV/AIDS in MSMs in China is 28% [6]. PrEP is a prominently effective prevention strategy for preventing HIV infection [40,46], and the medication adherence of PrEP users is an essential factor in the effectiveness of PrEP [10]. Therefore, HIV-preventive behavior was represented by the medication adherence rate of PrEP.

### 5.1. Self-Efficacy

Self-efficacy directly and positively affected HIV prevention behaviors. Similar results were obtained in previous studies; that is, self-efficacy is a critical variable in predicting HIV prevention behaviors in MSMs [38,39,40,47,48]. Self-efficacy means confidence in the ability to adhere to PrEP [41] and is the key mediating component between perceived benefits or barriers and protective behaviors [28,49] (Figure 2). In other words, MSMs with high self-efficacy have a high level of ability to eliminate perceived barriers and may thus have a high promotion effect on prevention behaviors. We found that self-efficacy directly and positively affects health behavior. Therefore, we believe that improving self-efficacy can significantly enhance the behavior against HIV infection in MSMs in western China.

### 5.2. Perceived Barriers

Perceived barriers refer to obstacles to HIV protective behavior in this study. Our study showed that perceived barriers negatively affect PrEP self-efficacy and HIV prevention behavior among MSM. In this study, the direct effect of perceived barriers on HIV protective behavior is not significant, but the direct effect of perceived barriers on self-efficacy is significant. These results can also be found in previous studies [23,28]. Accordingly, mediating effect of self-efficacy in the relation of perceived barriers with the HIV protective behavior. In other words, HIV-negative MSMs by breaking through psychological perceived barriers (such as peer pressure, stigma, drug efficacy, etc.) to reduce HIV prevention measures, showing that MSMs’ perceived barriers may be an essential factor in predicting their HIV prevention behavior indirectly. Therefore, relevant departments (Centers for Disease Control and Prevention, China Association for the prevention of AIDS) should increase the publicity of HIV/AIDS disease prevention, control knowledge, and introduce safe and effective HIV prevention drugs for MSMs to eliminate psychological obstacles to improve the practical improvement of HIV prevention behavior.

### 5.3. Perceived Benefits

The benefits of PrEP were perceived by the MSMs to be the physical and psychological benefits. We found that perceived benefits are significant factors associated with the HIV-preventive behaviors of the MSMs. This result is consistent with the results of some studies [35,36,38] but contrary to those of other studies [15,28].

Similar to perceived barriers, self-efficacy has a mediating effect on the relationship of perceived benefits with protective behavior against HIV. Therefore, high perceived benefits will likely promote HIV-preventive behavior or increase PrEP adherence rate. Hence, the government, organizations, and the media should emphasize the benefits of activities against HIV to reduce HIV infections in MSMs in China.

### 5.4. Perceived Susceptibility

HIV-preventive behavior was remarkably positively affected by perceived susceptibility. This result is consistent with previous studies [22,50]. Perceived susceptibility is defined as MSM’s subjective sense of threat from HIV infection, also a subjective assessment of the possibility of HIV infection among them. If MSMs think they are at a high risk of infection when engaging in unsafe sexual activities, such as condom-free intercourse, their perceived susceptibility will increase and their HIV-protective behaviors will increase accordingly. We found that perceived susceptibility had a positive impact on protection behavior. However, perceived susceptibility is not the main predictor of changes in behavior in the HBM [51], as also indicated by our results. This finding may be explained by the fact that perceiving susceptibility may play different roles in different populations or changes in behavior. In terms of disease prevention and control behavior, people will take preventive measures to prevent diseases because they feel they are at risk [33].

### 5.5. Perceived Severity

Perceived severity was defined as how serious an MSM assesses HIV infection. The study showed that perceived severity did not significantly affect the MSMs’ preventive behavior. The result is consistent with previous research [23]. HIV/AIDS is a sexually transmitted disease that cannot be completely cured, and MSMs compose a high-risk population for HIV infection cases. Hence, MSMs commonly have a high level of perception of the severity of HIV [31]; that is, a common belief has been reached. According to our study, this belief leads to the fact that perceived severity cannot be used in identifying differences in HIV prevention behavior among MSMs, and it is an unsuitable factor for predicting preventive behavior through perceived susceptibility.

### 5.6. Cues to Action

Cues to action mainly consist of external cues (whether to accept the reminder) and internal cues (whether to receive related knowledge) that can predict prevention behavior in MSMs. The results revealed that cues to action are positively related to MSMs’ behaviors that prevent HIV. Therefore, MSMs’ behavior for HIV prevention determines whether they will receive medication reminders or information. This finding is consistent with similar studies [42]. According to the knowledge, attitude, and practice theory, knowledge and attitude are crucial determinants of an individual’s ability to form certain health behaviors [52]. We found that medication reminders and information, cues to action, and interventions that promote preventive behavior improve actual preventive behavior and PrEP adherence. Cues to action are fundamental elements in MSMs’ health behavior enhancement theoretically. Therefore, cues to action are indispensable in predicting MSMs’ health behavior.

## 6. Conclusions

The HBM framework is acceptable for predicting MSMs’ HIV-preventive behaviors. In western China, medication self-efficacy, perceived susceptibility, and cues to action structures are predictors of MSMs’ HIV-preventive behavior. Moreover, medication self-efficacy was the most critical predictor of preventive behavior. Behavior interventions that help in overcoming psychological and social barriers and improving medication self-efficacy are crucial to improving preventive behavior efficiency or PrEP adherence.

## Figures and Tables

**Figure 1 ijerph-19-10185-f001:**
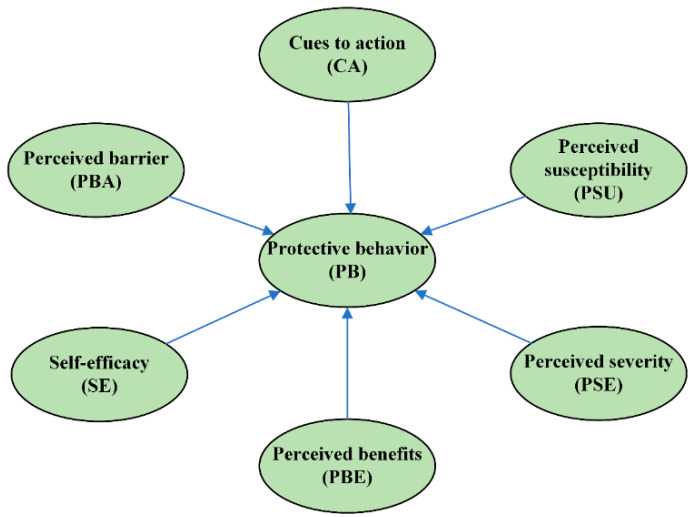
The HBM framework.

**Figure 2 ijerph-19-10185-f002:**
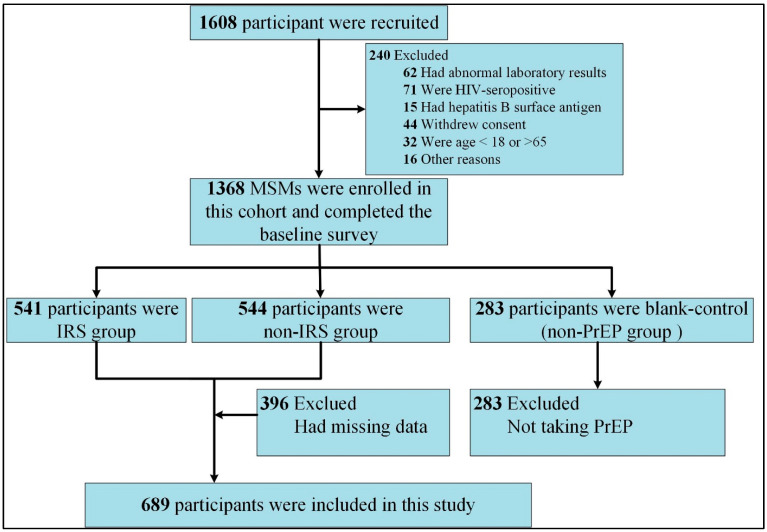
Flowchart of data selection: IRS group, taking PrEP and receiving reminder service from the IRS; non-IRS group, taking PrEP without receiving reminder service; non-PrEP group, neither taking PrEP drugs nor receiving reminder services.

**Figure 3 ijerph-19-10185-f003:**
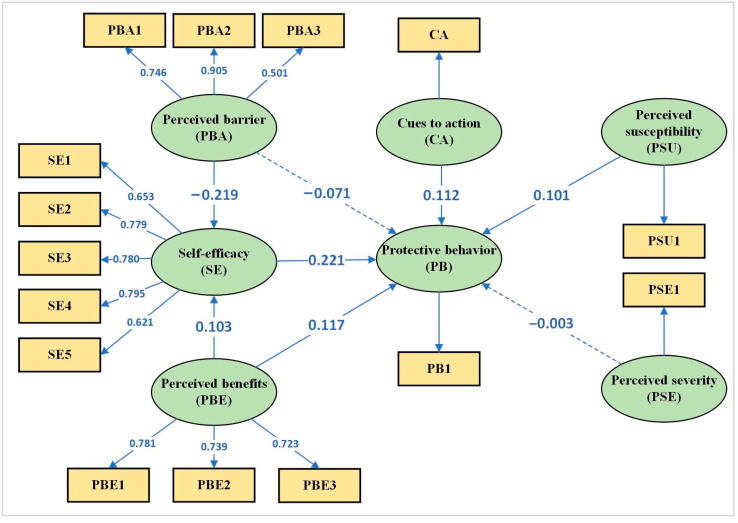
Structural equation model of direct and indirect pathways. Dashed lines indicate a non-significant effect.

**Table 1 ijerph-19-10185-t001:** Survey questions, value assignment, path coefficients, and factor loadings.

Constructs	Value Assignment	Factor Loading
**PSU** (β = 0.101 **)		
PSU1. How likely do you think you are of AIDS?	1 = Very small, 2 = Small, 3 = Average, 4 = Large, 5 = Very large	1.000 ^a^
**PSE** (β = −0.003)		
PSE1. What do you know about the severity of AIDS?	1 = Not serious at all, 2 = Not too serious, 3 = Somewhat serious, 4 = Serious, 5 = Very serious	1.000 ^a^
**PBE** (β = 0.117 **) (α = 0.791)		
PBE1. I think PrEP makes me safer and prevents AIDS.	1 = Do not agree at all, 2 = Agree a little, 3 = Agree to some extent, 4 = Mostly agree, 5 = Always	0.781 **
PBE2. I think PrEP makes me feel less afraid of AIDS.		0.739 **
PBE3. I can remember to take medicine on time.		0.723 **
**PBA** (β = −0.071) (α = 0.805)		
PBA1. I worry about homosexual partners learning that I am taking medication.	1 = Not at all, 2 = A little, 3 = Somewhat, 4 = Mostly, 5 = Always	0.746 **
PBA2. I worry that others will discriminate against me when they learn I am on medication.		0.905 **
PBA3. I worry about the side effects of the medicine.		0.501 **
**SE** (β = 0.221 **) (α = 0.802)		
SE1. When PrEP is not immediately available.	1 = Not confident at all, 2 = Not confident, 3 = Comparatively confident, 4 = Very confident, 5 = Completely confident	0.653 **
SE2. When your sex partner is upset about the PrEP.		0.799 **
SE3. When PrEP is too much trouble.		0.780 **
SE4. When your partner is angry about using PrEP.		0.795 **
SE5. When you have used other protective measures, such as condoms.		0.621 **
**CA** (β = 0.112 **)		1.000 ^a^
CA1. Have you gotten reminders from the intelligent reminder system?	0 = No, 1 = Yes	
CA2. Have you been tested for HIV/AIDS?		
CA3. Have you taken the initiative to receive free counseling on AIDS?		
**PB**		
PB1. PrEP adherence rate.	Continuous value from 0 to 1	1.000 ^a^

**Notes:**^a^ means fixed path coefficient. ** indicates statistical significance with *p* < 0.01; β indicates path coefficients among latent variables; α indicates Cronbach’s α. Abbreviations: PB, prevention behavior; PSU, perceived susceptibility; PSE, perceived severity; PBE, perceived benefits; PBA, perceived barriers; SE, self-efficacy; CA, cues to action.

**Table 2 ijerph-19-10185-t002:** The results of the latent factors correlation matrix.

Construct	PSU	PSE	PBE	PBA	SE	CA	PB
Perceived susceptibility (PSU)	1.000						
Perceived severity (PSE)	−0.005	1.000					
Perceived benefits (PBE)	0.020	−0.014	1.000				
Perceived barriers (PBA)	0.036	0.150 **	−0.014	1.000			
Self-efficacy (SE)	−0.028	0.001	0.195 **	−0.162 **	1.000		
Cues to action (CA)	0.089 **	0.030	0.087 **	−0.032	−0.006	1.000	
Preventive behavior (PB)	0.065 **	−0.039	0.151 **	−0.118 **	0.257 **	0.127 **	1.000

**Notes**: ** indicate statistical significance with *p* < 0.01.

**Table 3 ijerph-19-10185-t003:** Fit Indices.

Fit Index	χ^2^/df	GFI	NFI	CFI	RMSEA
Measurement model	1.954	0.982	0.965	0.976	0.037
Recommended value	<3	>0.9	>0.9	>0.94	<0.09

**Abbreviations**: GFI, goodness of fit index; NFI, normed fit index; CFI, comparative fit index; RMSEA, root mean-squared error of approximation.

## Data Availability

The datasets involved in the current study are not publicly available due to privacy but are available from the author on reasonable request.

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
