# Peer review of "The Influencing Factors of HIV-Preventive Behavior Based on Health Belief Model among HIV-Negative MSMs in Western China: A Structural Equation Modeling Analysis"

_ijerph, 2022, doi:10.3390/ijerph191610185_

Round 1
Reviewer 1 Report
In the introduction please provide more epidemiological data about HIV in China and explain why you focused on western China
Line 106 IRS – when you use abbreviation for the first time – please explain what does it mean
Line 109 – please explain: the group was between 18 and 65 or only in 18 and 65 age, it can be misunderstood
Line 110 – “one or more male partners one month before the trial” – please specify the exact timeframe
Line 117- „HIV negative after taking the 117 drug for one month” – please explain
Figure 2 explanation needs one more paragraph to be described, it doesn't match in the line 118
Author Response
We appreciate your kind suggestions and we have amended the manuscript accordingly. In addition, the followings are answers for your questions:
- In the introduction please provide more epidemiological data about HIV in China and explain why you focused on western China.
Response: Thanks for your valuable suggestion. We have added the epidemiological data description in our revised manuscript (Line 27-30, Page 1; Line 32-35, Page 1;).
There are two main reasons for this study to focus on Western China: (1) Western China has the highest number of HIV/AIDS infections in China [1-3], the government strongly supports the development of HIV/AIDS-related research in western China. (2) The scope of our research is limited to the western part of China. Most of our previous studies, we conducted, were concentrated in the western China, which provided an implementable basis for this study.
- Line 106: IRS - when you use abbreviation for the first time – please explain what does it mean.
Response: Thanks for your valuable suggestion. IRS refers to intelligent reminder system that is a timed reminder service for participants to take PrEP [4]. We added the corresponding abbreviation after “intelligent reminder system” in our revised manuscript (Line 103, Page 3; Line 128-130, Page 4).
- Line 109: please explain: the group was between 18 and 65 or only in 18 and 65 age, it can be misunderstood.
Response: Thanks for your valuable suggestion. Our study population ranged in age from 18 to 65 years. We modified the English expression in our revised manuscript (Line 111, Page 3).
- Line 110: “one or more male partners one month before the trial” – please specify the exact timeframe.
Response: Thanks for your valuable suggestion. The participants were asked to fill out a screening questionnaire during the screening phase, the question “How many male sexual partners have you had sex with, in the last month?” was used to assess participants' sexual partners one month before the screening phase. We only included those participants that if who had one or more male sexual partners one month before the screening phase. This inclusion criterion was used in our previous studies [5,6]. We added a more accurate timeframe to the revised manuscript as you suggested (Line 111-112, Page 3).
- Line 117: “HIV-negative after taking the drug for one month” – please explain.
Response: Thanks for your valuable suggestion. We have gone through a rigorous review, including the project plan and the data collation process, and found a mistake in the manuscript writing process. This exclusion criterion was not applied during the project execution and data consolidation process. We have deleted the mistake from the manuscript (Line 120, Page3).
- Line 118: Figure 2 explanation needs one more paragraph to be described, it doesn't match in the line 118.
Response: Thanks for your valuable suggestion. As seen in the response to question 5. we made a mistake in writing the manuscript, which caused 118 lines in the original manuscript to not match figure 2. We have deleted the mistake from the manuscript (Line 120, Page3).
We hereby resubmit the revised manuscript and hope that all corrections are satisfactory. The attached document is our revised manuscript. Please feel free to contact us with any questions and we look forward to your decision.
References:
- Ye, L.; Wei, S.; Zou, Y.; Yang, X.; Abdullah, A.S.; Zhong, X.; Ruan, Y.; Lin, X.; Li, M.; Wu, D.; et al. HIV pre-exposure prophy-laxis interest among female sex workers in Guangxi, China. PLoS One 2014, 9, e86200, doi:10.1371/journal.pone.0086200.
- Jia-jin, H.; Lu, Y.; Chao, W. Temporal-spatial distribution of AIDS epidemic in China, 2010-2019. CHINESE JOURNAL OF DISEASE CONTROL & PREVENTION 2022, 26, 541, doi:10.16462/j.cnki.zhjbkz.2022.05.009.
- Chen, F.F.; Guo, W.; Qin, Q.Q.; Cai, C.; Cui, Y. Spatial-temporal distribution of newly detected HIV/AIDS cases among aged 15 years or older women in China, 2010-2016. Zhonghua Liu Xing Bing Xue Za Zhi 2018, 39, 739-744, doi:10.3760/cma.j.issn.0254-6450.2018.06.009.
- Ma, Y.; Zhong, X.; Lin, B.; He, W. Factors Influencing the Intention of MSM to Use the PrEP Intelligent Reminder System. Risk Manag Healthc Policy 2021, 14, 4739-4748, doi:10.2147/RMHP.S337287.
- Liu, J.; Zhong, X.; Lu, Z.; Peng, B.; Zhang, Y.; Liang, H.; Dai, J.; Zhang, J.; Huang, A. Anxiety and Depression Associated with Anal Sexual Practices among HIV-Negative Men Who Have Sex with Men in Western China. Int J Environ Res Public Health 2020, 17, doi:10.3390/ijerph17020464.
- Liu, J.; Deng, R.; Lin, B.; Pan, H.; Gao, Y.; Dai, J.; Liang, H.; Huang, A.; Zhong, X. Risk Management on Pre-Exposure Prophylaxis Adherence of Men Who Have Sex with Multiple Men: A Multicenter Prospective Cohort Study. Risk Manag Healthc Policy 2021, 14, 1749-1761, doi:10.2147/RMHP.S295114.
Reviewer 2 Report
IJERPH – 1834626 – v1 The influencing factors of HIV preventive behavior based on Health belief Model among HIV-negative MSM in western Chine: A structural equation modeling analysis
|
|
|
|
|
This submission examines pre-exposure prophylaxis use to prevent HIV transmission among males who have sex with men residing in western China. Prophylaxis use is examined in relation to the health Behavioral Model (HBM). The authors subscribe a specific hypothesis to each of the six components of the HBM. A review of the English language is needed.
The authors indicate that volunteer participants from an established cohort were randomly divided into three groups. However, the significance of the three groups is not explained. The methods section tends to state the data that was collected but does not actually describe how the data was acquired. Was the demographic information drawn from the cohort study? If so, then briefly state the procedure (self or interviewer administered questionnaire, tec.) and cite that study. Were validated question used and how were the HBM questions assessed? No information is given regarding how the “medication” was acquired and usage assessed by participants, only that a tally was completed to measure “protective behavior”.
Table 3 appears to provide survey questions. Six of these questions specifically inquire about Acquired Immune Deficiency Syndrome (AIDS) which is significantly different than Human Immunodeficiency Virus (HIV) which can lead to AIDS but is not AIDS. These questions are concerning as the authors are aiming to determine the association of prophylaxis use to protect from HIV, but are asking questions about AIDS. No information is provided indicating the level of understanding of HIV and potential risk of AIDS among participants.
This manuscript is an interesting area of work but currently lacks a clear description of how the study was completed and justification of using AIDS risk as a proxy for HIV-related behaviors. Lacking the detailed foundation, it not currently possible to determine the significance of stated results.
Specific notes:
Introduction
Last sentence of first paragraph is confusing. Are you saying that 25% of new HIV infections are among MSM or that of among those that are living with HIV infection in China, 25% are MSM?
Table 1 Writing out each HBM element in the column headings followed be the abbreviation that can be used for the row headings would help clarify Table 1.
Figures require more descriptive information than that currently provided.
Author Response
We appreciate your kind suggestions and we have amended the manuscript accordingly. In addition, the followings are answers for your questions:
- The authors subscribe a specific hypothesis to each of the six components of the HBM. A review of the English language is needed.
Response: Thanks for your valuable suggestion. We have corrected corresponding spelling or grammar errors. This manuscript has been edited by International Science Editing (http://www.kgsupport.com).
- The authors indicate that volunteer participants from an established cohort were randomly divided into three groups. However, the significance of the three groups is not explained.
Response: Thanks for your valuable suggestion. IRS refers to an intelligent reminder system that is a timed reminder service for participants to take PrEP [1].
IRS group: taking PrEP and receiving reminder service from the IRS.
Non-IRS group: taking PrEP without receiving reminder service.
Non-PrEP group: neither take PrEP drugs nor receive reminder services.
Since PrEP adherence rates measured protective behavior, the blank control group was not included in our study (Figure 2). The only difference between the IRS group and the non-IRS group was whether or not they received reminder service from IRS.
As your suggestion, we described the IRS, non-IRS, and non-PrEP groups in detail in the footnote in Figure 2 (Line 128-130, Page 4).
- The methods section tends to state the data that was collected but does not actually describe how the data was acquired. Was the demographic information drawn from the cohort study? If so, then briefly state the procedure (self or interviewer administered questionnaire, tec.) and cite that study.
Response: Thanks for your valuable suggestion. All the data in this study were obtained from this cohort study (National Science and Technology Major Project in Western China). We selected survey data for the first follow-up period of this cohort study, since HBM-related questions were only included in the questionnaire for the first follow-up period. As your suggestions, we added a description of the data collection procedure at the end of Chapter 3.1. Participants and procedure and referenced the articles produced by the project (Line 120-124, Page 3).
- Were validated question used and how were the HBM questions assessed?
Response: Thanks for your valuable suggestion. HBM includes perceived susceptibility, severity, benefits, barriers, cues to action, and protective behaviors. Since the problems with measuring HBM in different studies are not identical, the questions we chose to e HBM in this study were all validated by previous research [2,3], besides PrEP adherence rate. We have slightly adjusted the structure of Table 3 to show more clearly the value assignment for each problem (Line 208-209, Page 6-7).
- No information is given regarding how the “medication” was acquired and usage assessed by participants, only that a tally was completed to measure “protective behavior”.
Response: Thanks for your valuable suggestion. In this study, participants gained free PrEP medication at the project's clinical research center every follow-up period (the cost of the PrEP is fully covered by the project). At the same time as the PrEP was distributed, participants were asked to complete a medication questionnaire. The questionnaire asked participants how many times they had taken their medication (PrEP as prescribed) in both follow-up period (baseline to first follow-up, in this study), and how many times they had missed it (e.g., due to forgetting, do not want to take PrEP). We think of taking PrEP as a protective behavior to prevent HIV/AIDS, and adherence to PrEP is an objective measure of protective behavior. We think it is reasonable to use medication compliance rate to measure protective behavior.
As your suggestions, we added brief information about subject access to the drug to the revised manuscript (Line 141-143, Page 4).
- Table 3 appears to provide survey questions. Six of these questions specifically inquire about Acquired Immune Deficiency Syndrome (AIDS) which is significantly different than Human Immunodeficiency Virus (HIV) which can lead to AIDS but is not AIDS. These questions are concerning as the authors are aiming to determine the association of prophylaxis use to protect from HIV, but are asking questions about AIDS. No information is provided indicating the level of understanding of HIV and potential risk of AIDS among participants.
Response: Thanks for your valuable suggestion. We also considered this problem when we started our research. However, we still choose to use "AIDS" instead of "HIV infection" for the following two reasons: (1) There are many types of HIV virus (e.g., HIV-1, HIV-2), and each type contains many subtypes (e.g., A, B, C, D, etc.). But in our study, we did not distinguish between multiple classes of HIV. So, to avoid ambiguity, we used AIDS instead of HIV infection in the questionnaire. (2) AIDS is caused by HIV infection, AIDS is a direct manifestation of HIV infection. Most people percept to AIDS is more accurate and direct than HIV. Therefore, we indirectly reflect the perception of HIV infection through the perception of AIDS.
Since our research is mainly in the fields related to MSM and PrEP, we did not research the participants' understanding of HIV/AIDS. But, a large number of studies have shown that people in China generally have a low level of understanding of HIV/AIDS [4-6].
- Last sentence of first paragraph is confusing. Are you saying that 25% of new HIV infections are among MSM or that of among those that are living with HIV infection in China, 25% are MSM?
Response: Thanks for your valuable suggestions. Since we use improper grammar, it leads to ambiguity emerged. This sentence means MSM accounts for about 25% of new HIV infections in China. We have modified the original manuscript according to your suggestions (Line 31-33, Page 1).
- Table 1 Writing out each HBM element in the column headings followed be the abbreviation that can be used for the row headings would help clarify Table 1.
Response: Thanks for your valuable suggestions. As your suggestions, we made minor adjustments to Table 1 (Line 183, Page 5).
- Figures require more descriptive information than that currently provided.
Response: Thanks for your valuable suggestions. As your suggestions, we modified figure 2 (Line 127, Page 4) and added the description for figure 2 (Line 128-130, Page 4).
We hereby resubmit the revised manuscript and hope that all corrections are satisfactory. Please feel free to contact us with any questions and we look forward to your decision.
Reference
- Ma, Y.; Zhong, X.; Lin, B.; He, W. Factors Influencing the Intention of MSM to Use the PrEP Intelligent Reminder System. Risk Manag Healthc Policy 2021, 14, 4739-4748, doi:10.2147/RMHP.S337287.
- Liu, J.; Deng, R.; Lin, B.; Pan, H.; Gao, Y.; Dai, J.; Liang, H.; Huang, A.; Zhong, X. Risk Management on Pre-Exposure Prophylaxis Adherence of Men Who Have Sex with Multiple Men: A Multicenter Prospective Cohort Study. Risk Manag Healthc Policy 2021, 14, 1749-1761, doi:10.2147/RMHP.S295114.
- Huang, S.T.; Huang, J.H.; Chu, J.H. Health Beliefs Linked to HIV Pre-Exposure Prophylaxis Use Intention Among Young Men Who Have Sex with Men in Taiwan. AIDS Patient Care STDS 2021, 35, 474-480, doi:10.1089/apc.2021.0146.
- Zhao, C.; Xiang, X.; Guo, K.; Liu, X.; Cao, J.; Li, C.; Wang, Q. Study on Knowledge, Attitudes and Behaviours About AIDS Among Undergraduates of China. Curr HIV Res 2021, 19, 304-310, doi:10.2174/1570162X18666201218121200.
- Lai, J.; Pan, P.; Lin, Y.; Ye, L.; Xie, L.; Xie, Y.; Liang, B.; Zheng, F.; Chen, R.; Wen, L.; et al. A Survey on HIV/AIDS-Related Knowledge, Attitudes, Risk Behaviors, and Characteristics of Men Who Have Sex with Men among University Students in Guangxi, China. Biomed Res Int 2020, 2020, 7857231, doi:10.1155/2020/7857231.
- Chen, M.; Liao, Y.; Liu, J.; Fang, W.; Hong, N.; Ye, X.; Li, J.; Tang, Q.; Pan, W.; Liao, W. Comparison of Sexual Knowledge, Attitude, and Behavior between Female Chinese College Students from Urban Areas and Rural Areas: A Hidden Challenge for HIV/AIDS Control in China. Biomed Res Int 2016, 2016, 8175921, doi:10.1155/2016/8175921.

Round 2
Reviewer 1 Report
the authors' explanations and corrections are satisfactory